# Human Papillomavirus-Encoded microRNAs as Regulators of Human Gene Expression in Anal Squamous Cell Carcinoma: A Meta-Transcriptomics Study

**DOI:** 10.3390/ncrna11030043

**Published:** 2025-06-09

**Authors:** Daniel J. García, Marco A. Pulpillo-Berrocal, José L. Ruiz, Eduardo Andrés-León, Laura C. Terrón-Camero

**Affiliations:** 1Institute of Parasitology and Biomedicine “López-Neyra” (IPBLN), Spanish National Research Council (CSIC), 18016 Granada, Spain; daniel.garcia@csic.es (D.J.G.); marcoapolo@ipb.csic.es (M.A.P.-B.); joseluis.ruiz@csic.es (J.L.R.); 2Functional Genomics Center Zurich, ETH Zurich, University of Zurich, 8057 Zurich, Switzerland

**Keywords:** de novo miRNAs, viral miRNAs, anal squamous cell carcinoma, squamous intraepithelial lesion, human *papillomavirus*, transcriptomics, target prediction, oncomiR

## Abstract

**Introduction:** Anal squamous cell carcinoma (ASCC) is a rare but increasingly common gastrointestinal malignancy, mainly associated with oncogenic human papillomaviruses (HPVs). The role of non-coding RNAs (ncRNAs) in tumorigenesis is recognized, but the impact of viral ncRNAs on host gene expression remains unclear. **Methods:** We re-analyzed total RNA-Seq data from 70 anal biopsies: 31 low-grade squamous intraepithelial lesions (LGSIL), 16 high-grade SIL (HGSIL), and 23 ASCC cases. Microbial composition was assessed taxonomically. Novel viral miRNAs were predicted using vsRNAfinder and linked to host targets using TargetScan and expression correlation analyses. **Results:** Microbial profiling revealed significant differences in abundance, with Alphapapillomaviruses types 9, 10, and 14 enriched across lesion grades. We identified 90 novel viral miRNAs and 177 significant anti-correlated miRNA–mRNA interactions. Target genes were enriched in pathways related to cell cycle, epithelial–mesenchymal transition, lipid metabolism, immune modulation, and viral replication. **Discussion:** Our findings suggest that HPV-derived miRNAs, including those from low-risk types, may contribute to neoplastic transformation by modulating host regulatory networks. **Conclusion:** This study highlights viral miRNAs as potential drivers of HPV-related anal cancer and supports their utility as early biomarkers and therapeutic targets in ASCC.

## 1. Introduction

The vast majority of the human genome (~98%) encodes RNAs that do not undergo translation into proteins at the ribosomes, collectively referred to as non-coding RNAs (ncRNAs) [1,2]. Since their identification as ‘junk RNA’, ncRNAs have gained attention for their crucial biological roles, driving a fundamental paradigm in RNA research. Although initially considered intermediaries in protein synthesis or degradation products of RNA metabolism, they are now understood as key regulatory molecules that influence genome organization and control gene expression [3].

Among non-coding RNAs (ncRNAs), microRNAs (miRNAs) and long non-coding RNAs (lncRNAs) have garnered particular interest due to their relevant roles in a plethora of essential biological processes. miRNAs are small (~22 nucleotides), evolutionarily conserved molecules that primarily function in post-transcriptional gene silencing-related cell differentiation and proliferation, embryonic development, and homeostasis [4,5]. In contrast, lncRNAs are typically defined as transcripts longer than 200 nucleotides and are involved in a wide array of regulatory mechanisms, including chromatin remodeling, transcriptional control, and RNA processing [6]. The functional versatility of these ncRNAs underscores their significance in both normal physiology and disease [7].

miRNAs modulate gene expression by interacting with the 3′ untranslated regions (UTRs) of target mRNAs, usually suppressing translation or influencing stability and degradation [8,9]. Recent findings have revealed that many miRNAs map to exons of protein-coding genes (PCG), further increasing the complexity of these functional non-coding elements in the genomic regulatory organization [9]. In addition to modulating gene expression, miRNAs can also participate in intercellular communication, being secreted in extracellular vesicles [10] or serving as ligands to activate TLR signaling pathways, thereby inducing an immune response [11,12]. Numerous studies have demonstrated that dysregulated microRNA (miRNA) expression is intricately linked to the pathophysiology of a wide range of cellular processes. These include cardiovascular diseases [13], neurodegenerative disorders [14], immune system dysfunctions [15,16,17], and inflammatory conditions [18]. Notably, miRNA deregulation has been most extensively studied in the context of cancer, where it plays a critical role in tumor initiation, progression, and metastasis [19]. Additional evidence further supports the involvement of altered miRNA expression in other pathological states [18], highlighting their broad relevance in human disease. Since the early 2000s, accumulating evidence has underscored the role of dysregulated miRNAs, specifically functioning as oncogenes or tumor suppressors, in tumorigenesis [19]. A major seminal discovery in this field was the identification of the deletion of miR-5 and miR-6 in patients with chronic lymphocytic leukemia [18]. Another study investigated the let-7 family, which typically acts as a tumor suppressor by limiting tumor progression and metastasis [20]. Overall, miRNAs have emerged not only as valuable biomarkers for cancer development but also as important regulators of treatment responses, influencing drug resistance and therapeutic efficacy [21,22,23].

In recent decades, extensive research on various diseases, including cancer, diabetes, and neurological disorders [24], has also highlighted the functional importance and key roles of the microbiota, for example, maintaining homeostasis and modulating host gene expression [25]. Over the past few years, ncRNAs have also been identified as crucial regulators of the dynamic interplay between pathogenic microorganisms and their hosts [26]. Growing evidence suggest that microbial ncRNAs, such as miRNAs and lncRNAs, contribute to the alteration of host RNA expression profiles, either by producing their own ncRNAs to be incorporated by the host or by modulating host ncRNAs [27,28]. Several key processes in this context of host–microbiota interactions can be affected, including host inflammatory responses, cell invasion, tissue remodeling, and both innate and adaptive immunity [26,29,30]. In particular, viral miRNAs may contribute to regulating both pathogen and host gene expression, for example, to control the transition between viral phases, facilitate viral replication by enhancing cell survival, or influence immune responses. Thus, viral proteins and miRNAs could establish a favorable host environment for viral persistence [30]. For instance, miR-122 binding at the 5′ non-coding region of the hepatitis C virus is essential for viral RNA stability and increased infection rates [31,32]. In that sense, miR-122 binding has been reported to induce a sponge effect, sequestering free miR-122 at the site of infection. Consequently, the reduction in overall miR-122 levels potentially disrupts liver homeostasis and increases the risk of hepatocarcinoma [33]. In the same tumor context, Epstein–Barr virus (EBV) miRNAs have been observed to reduce the expression of the transcriptional repressor BCL6 in diffuse large B-cell lymphoma, which may be necessary to promote the survival of EBV-positive neoplastic cells [34]. Another strategy employed by viruses is the mimicry of host miRNAs. By replicating their sequence and structure, viral miRNAs can exploit the host gene regulatory machinery, controlling gene networks and facilitating disease progression and the establishment of chronic infections [35,36]. Viral miRNA mimicry typically retains only the similarity of the seed sequence, so that the set of potential target genes, known as the targetome, is not completely identical to that of the host miRNA [36]. For instance, the Kaposi’s sarcoma-associated herpesvirus (KSHV) encodes miR-K6-5p as a mimic of the tumor-suppressive cellular miR-15/16 miRNA family, so it can fine-tune its viral oncogene expression to prevent severe pathogenesis in a healthy host [37].

Similar to miRNAs, the regulatory function of lncRNAs in viral infection has been recognized for decades, with accumulating evidence demonstrating their involvement in various stages of the viral life cycle, resistance to antiviral immune response, and viral pathogenicity [38]. The PAN (polyadenylated nuclear RNA) lncRNA is implicated in multiple key stages of the viral life cycle, including the initiation of viral replication, repression of host immune regulator expression, regulation of cellular pluripotency, and mediation of host–pathogen interactions [38,39,40]. In addition to regulating gene expression, lncRNAs can directly interact with proteins, modulating viral processes and host–virus dynamics. For example, the lncRNA called low-molecular-weight tristeza 1 (LMT1) interacts with the Citrus Tristeza Virus (CTV) p33 protein, contributing to the maintenance of viral persistence throughout the life of the host organism [41]. Regarding cancer, EBV expresses lncRNAs that modulate host gene expression to facilitate infection and may contribute to tumor development. For instance, BART lncRNAs influence the regulation of genes associated with inflammatory responses, cell adhesion, and metastasis [42]. Similarly, it has been also demonstrated that the human cytomegalovirus lncRNA4.9 induces the rapid growth of triple-negative breast tumors in NOD/SCID gamma mice [43]. Beyond ncRNAs and miRNAs, there are other classes of viral non-coding RNA that enable viral entities to manipulate host cellular dynamics to facilitate their life cycle, such as the small nuclear ncRNAs (snRNAs) in γ-herpesviruses and the virus-associated (VA) ncRNAs in adenoviruses [44].

Due to the potential viral etiology of certain cancers, the analysis of miRNA profiles could be essential for identifying novel therapeutic targets [45]. Anal squamous cell carcinoma (ASCC) is one such cancer associated with viral presence. It is driven by infection with oncogenic human *Alphapapillomavirus* strains [46], and it accounts for up to 3% of gastrointestinal tumors [47]. In recent years, the global incidence of ASCC has increased, mainly among men who have sex with men and people living with HIV [48]. High-risk human papillomaviruses 16 (HPV) and 18 are responsible for the development of ASCC in approximately 80–85% of cases [49]. According to the grade of neoplasia, squamous intraepithelial lesions (SILs) typically precede the onset of ASCC and are categorized into low-grade SIL (LGSIL) and high-grade SIL (HGSIL) subtypes [50]. Infections with low-risk HPV subtypes, such as HPV6 and HPV11, are mainly linked to these SILs, including genital warts or condyloma acuminata [51,52]. Histology and immunohistochemistry remain the primary methods for diagnosing the SIL subtype, although there is still considerable variability in accurately defining the grade of the lesion [53,54]. Previous research has demonstrated that ASCC cases associated with HPV exhibit distinct clinicopathological covariates compared to those that are HPV-negative. Specifically, HPV-negative ASCCs are more often characterized by keratinization, lack of p16 immunoreactivity, and poorer survival rates compared to HPV-positive tumors [55,56]. A weakened immune system is another significant risk factor for ASCC, which is often found in individuals with HIV or organ transplant recipients who require long-term immunosuppressive therapy [57,58]. In addition, growing evidence suggests that microbial communities, including HPV, play a role in tumor development, particularly in HPV-associated malignancies [59,60]. Since preventing the progression of SILs could significantly reduce the risk of ASCC, identifying viral miRNAs in patient samples may provide valuable insights into early detection of the disease. Given the lack of reliable prognostic biomarkers, investigating transcriptomic alterations in the microbiome present in ASCC samples compared with SILs could be crucial for improving early diagnosis and risk stratification. The involvement of HPV16-encoded miRNAs, such as HPV16-miR-H1 and HPV16-miR-H6, in cervical cancer is well established, as they have been shown to enhance tumor progression by regulating enhancer activity and promoting cell migration [61]. However, little is known about the role of miRNAs in ASCC and their potential impact on tumor development.

Overall, although great efforts have been made to elucidate the functions of ncRNAs, the extent to which microbial organisms can regulate host gene expression through ncRNAs remains poorly understood. Considering the morphological and transcriptomic heterogeneity observed in different ASCC stages among patients and the established involvement of viruses in this malignancy, the detection of dysregulated viral miRNAs could open new avenues for improving clinical diagnosis and prognosis. By conforming to the FAIR data principles, in this study we leveraged already available total RNA datasets addressing different ASCC stages. Applying an up-to-date meta-transcriptomics workflow, our aim was to investigate and characterize the dysregulated viral miRNAs, together with their potential role in modulating host gene expression in a pathogenic context.

## 2. Results

### 2.1. Differential Microbial Abundance Across Precursor Lesions and ASCC

Our meta-transcriptomic pipeline enabled comprehensive profiling of all RNA species present in the samples analyzed by Lacunza et al. (2024) [62]. Out of these, ~1% were identified as non-human sequences and aligned to microorganisms’ genomes (see Methods). In total, 427 transcriptionally active microorganisms were identified in the patient samples. Among these, 88.60% belonged to bacteria, 5.00% to fungi, 2.14% to viruses, and 4.30% to archaea and protozoa. Next, we applied a novel computational approach to assess microbial differential abundance in four comparisons: (i) anal squamous cell carcinoma (ASCC) vs. high-grade squamous intraepithelial lesions (HGSILs), (ii) ASCC vs. low-grade squamous intraepithelial lesions (LGSILs), (iii) HGSILs vs. LGSILs, and (iv) a broader comparison of patients presenting ASCC vs. a grouping of high- and low-grade SILs (HGSILs + LGSILs; Appendix A).

When comparing ASCC vs. HGSILs, 85 differentially abundant microorganisms were detected, *Alphapapillomavirus* 10 (logFC = −7.05, FDR = 2.93 × 10^−4^) and *Alphapapillomavirus* 14 (logFC = −5.30, FDR = 1.31 × 10^−3^) exhibiting greater expression inhibition in ASCC. Between ASCC and LGSILs, 93 differentially abundant microorganisms were identified. In this comparison, *Alphapapillomavirus* 9 showed a higher abundance in ASCC samples (logFC = 5.91, FDR = 2.86 × 10^−4^), while *Alphapapillomavirus* 14 showed a lower abundance in ASCC (logFC = −4.09, FDR = 4.20 × 10^−3^). In addition, a significant decrease in the abundance of other *Alphapapillomaviruses* was observed, specifically *Alphapapillomavirus* 10 (logFC = −5.64, FDR = 9.35 × 10^−4^), i.e., a distinctive viral composition associated with ASCC. In the HGSIL vs. LGSIL comparison, we characterized 427 differentially abundant microorganisms (FDR < 0.05) corresponding to 20 microbial species (i.e., differentially abundant microorganisms), including bacteria and viruses. In this analysis, *Alphapapillomavirus* 9 was found to be significantly more abundant in the HGSIL samples against the LGSIL samples (logFC = 6.93, FDR = 1.68 × 10^−4^). Finally, we aimed to identify differences between patients with lesions and cancer. In the comparison between ASCC and LGSIL/HGSIL samples, we identified 47 differentially abundant microorganisms. Notably, *Alphapapillomavirus* 10 was significantly more abundant in LGSIL/HGSIL samples (logFC = –5.82, FDR = 1.20 × 10^−3^) (Figure 1; Appendix A).

### 2.2. Detection and Profiling of Viral miRNAs

Given the significant differential abundance of Alphapapillomavirus species 9, 10, and 14 observed in our analysis, we next investigated their potential impact on host transcriptomic regulation through viral miRNAs. These viral groups comprise multiple HPV genotypes associated with distinct pathological outcomes. Specifically, Alphapapillomavirus 9 includes high-risk genotypes such as HPV 16, 31, 33, 52, 58, and 67; Alphapapillomavirus 10 contains predominantly low-risk types, including HPV 6, 11, 13, 44, and 74; and Alphapapillomavirus 14 encompasses HPV 71, 90, and 106 [63]. This classification provided a framework for exploring the miRNA-mediated regulatory potential of individual strains within each viral group. Considering the limited information on miRNAs characterized in these HPVs of interest, we used their viral genomes to infer potential miRNAs (Appendix A). We predicted 90 significant viral miRNAs in this subset of Alphapapillomaviruses: 5 in HPV6, 32 in HPV11, 6 in HPV16, 1 in HPV33, 1 in HPV35, 15 in HPV39, 6 in HPV42, and 14 in HPV90. The sequences and details of the identified miRNAs are included in Appendix A.

To explore the potential regulatory roles of the newly identified miRNAs, we investigated their patterns of differential expression. For the comparisons between groups of patient samples depicted above revealed significant variations in viral miRNA expression only for HPV6, HPV11, and HPV90 (Figure 2). The highest number of differentially expressed miRNAs was observed in the comparisons of HGSIL vs. LGSIL (N = 36) and ASCC vs. HGSIL + LGSIL lesions (N = 37). In contrast, the comparisons of ASCC vs. HGSILs and ASCC vs. LGSILs resulted in a total of 27 and 26 differentially expressed miRNAs, respectively (Appendix A). In the case of HPV90, only the miRNA 3357-3373---AY057438.1 was differentially expressed, upregulated in the high-grade SIL samples for the HGSIL vs. LGSIL comparison (logFC = 1.85, FDR = 4.78 × 10−3) and downregulated in the cancerous samples for the ASCC vs. HGSIL comparison (logFC = −1.85, FDR = 0.03). Although Alphapapillomavirus 9 was the only virus found to be differentially abundant in the HGSIL vs. LGSIL comparison, no differentially expressed miRNAs associated with its constituent genotypes were detected. Consequently, this comparative was excluded from further analysis.

### 2.3. Prediction and Correlation of Viral miRNA Targets with Host Gene Expression

After identifying and characterizing the expression patterns of the viral miRNAs described above, we next investigated the differential transcriptional profiles of the human host that may be subject to regulation by these predicted viral miRNAs. To this end, we analyzed the human-derived fractions of the transcriptomic datasets. In the comparison among ASCC and HGSILs, 170 differentially expressed genes (FDR < 0.01) were identified, 43 downregulated and 127 upregulated (Figure 3A). Comparison between ASCC and LGSILs showed a higher number of affected genes, with 1927 differentially expressed genes, including 701 downregulated and 1226 upregulated genes (Figure 3B). In the last comparison grouping LGSILs and HGSILs into a single category versus ASCC, 1777 differentially expressed genes were detected, with 602 genes downregulated and 1175 upregulated (Figure 3C). These differentially expressed PCGs could be targets of the de novo miRNAs that we predicted above. Therefore, we next computed their potential interactions considering significant genes and HPV-miRNAs (FDR < 0.01 and FDR < 0.05, respectively; Appendix A).

The analysis of interactions between the identified miRNAs and differentially expressed PCGs yielded a variable number of potential targets across comparisons. In the ASCC vs. HGSIL comparison, 6407 potential target genes were identified, whereas in the ASCC vs. LGSIL comparison the number of targets reached 79,140 and in the comparison of ASCC vs. HGSILs + LGSILs a total of 95,036 targets were predicted. All identified interactions correspond to the 8-6mer type, a pairing pattern characterized by high affinity and specificity in miRNA binding to its target transcript (Appendix A).

In addition to the prediction of potential targets, an anti-correlation analysis was performed between the miRNAs and differentially expressed genes predicted to interact to investigate the potentially negative regulation of miRNAs on genes in the groups of interest (Figure 4). This approach allowed us to identify negative correlation profiles between miRNAs and their predicted target genes, which could be indicative of a post-transcriptional regulatory mechanism. To further validate and investigate this relationship in silico, we performed correlation analyses of the expression levels of the miRNAs and predicted target PCGs.

Notable differences were observed across the comparisons, reflecting varying degrees of miRNA–PCG regulatory interactions associated with lesion severity. The ASCC vs. HGSIL + LGSIL comparison showed the highest number of significant anti-correlations (177); in the ASCC vs. LGSIL comparison, 96 significant anti-correlations were identified; in contrast, the ASCC vs. HGSIL comparison revealed only 7 significant anti-correlations (Figure 4).

To examine in more detail the functional impact of these regulatory interactions, an enrichment analysis was performed on the differentially expressed PCGs potentially targeted by miRNAs. The comparison between ASCC and HGSILs potentially targeted by miRNAs identified 26 significant biological pathways, highlighting various key metabolic and cellular processes involved in the tumor progression (Figure 5A). These include several related to the cell cycle, such as meiotic nuclear division (*p* = 2.56 × 10^−3^) and the meiotic cell-cycle process (*p* = 3.06 × 10^−3^), indicating possible regulation of cell division in the early stages of the lesions. Additionally, several pathways associated with lipid metabolism were detected, such as regulation of the phosphatidylcholine metabolic process (*p* = 4.11 × 10^−3^), negative regulation of lipase activity (*p* = 6.56 × 10^−3^), and negative regulation of the fatty acid biosynthetic process (*p* = 6.56 × 10^−3^).

In the functional enrichment analysis of the differentially expressed miRNA-targeted PCGs in the comparison between ASCC and LGSILs, a total of 33 significant biological pathways were identified (Figure 5B). Among the most prominent pathways was the positive regulation of the Smoothened signaling pathway (*p* = 6.57 × 10^−4^), which is related to the Hedgehog signaling pathway, a key process in the regulation of cell development and proliferation. In addition, epithelial cell proliferation (*p* = 1.85 × 10^−3^) was the most enriched pathway, with eight distinct genes, closely followed by the regulation of the epithelial cell proliferation pathway (*p* = 1.19 × 10^−3^), which included six related genes. Other observed pathways, such as mesenchymal cell proliferation (*p* = 1.37 × 10^−3^) and the Wnt signaling pathway (*p* = 2.76 × 10^−2^), were identified during the enrichment analysis.

In the functional enrichment analysis of the differentially expressed miRNA-targeted PCGs in the comparison between ASCC and HGSILs + LGSILs, 46 significant biological pathways were identified (Figure 5C). Similar to the ASCC vs. LGSIL comparison, among the most prominent pathways were those associated with the cell cycle, such as the positive regulation of the meiotic cell cycle (*p* = 9.66 × 10^−6^) and meiotic nuclear division (*p* = 2.60 × 10^−4^), the Smoothened signaling pathway (*p* = 4.60 × 10^−4^), and mesenchymal cell proliferation (*p* = 7.50 × 10^−4^). Finally, other results include additional pathways related to viral processes (*p* = 2.55 × 10^−3^) and cell division (*p* = 1.95 × 10^−2^), highlighting potential mechanisms involved in disease progression (Appendix A).

## 3. Discussion

This study aimed to identify de novo viral miRNAs in previously published transcriptomic datasets derived from anal biopsy samples of patients with precursor lesions and ASCC [62]. To achieve this objective, a novel computational workflow was developed, and microbial abundance analysis was applied to the following sample group comparisons: (i) ASCC vs. HGSILs, (ii) ASCC vs. LGSILs, (iii) HGSILs vs. LGSILs, and (iv) ASCC vs. HGSILs + LGSILs. The number of differentially abundant microorganisms was higher in the ASCC vs. LGSIL comparison, while the lowest was observed in the HGSIL vs. LGSIL comparison. These results indicate a greater imbalance in microbial communities during carcinogenesis. Despite using a different methodology and experimental design to those of Lacunza et al., 2024 [62], our findings are consistent with their observations—notably, the high abundance of Alphapapillomaviruses 9, 10, and 14, which were among the most prevalent viral species identified across diagnostic groups in their study. These Alphapapillomavirus groups include different HPV strains with distinct carcinogenic potential. Alphapapillomavirus 9 is exclusively composed of high-risk HPV strains, including HPV-16, one of the most well-established oncogenic viruses [64]. In contrast, Alphapapillomaviruses 10 and 14 contain only low-risk HPV types, which are generally not associated with malignant progression. However, it is well described in the literature that numerous HPV types, including both high-risk and low-risk strains, have been implicated in carcinogenesis [65].

Since the discovery of the first viral miRNA in Epstein–Barr virus (EBV) in 2004, numerous others have been reported to play roles in various diseases, underscoring their potential involvement in viral pathogenesis and host regulation [61,66]. In this context, we aimed to investigate the role of viral miRNAs by characterizing the expression patterns of a newly predicted de novo miRNA dataset in order to assess their potential contribution to tumorigenesis during ASCC progression.

Previous studies have provided strong evidence that viral miRNAs, such as HPV16-miR-H1 and HPV16-miR-H6, alter the expression of host genes regulating the cell cycle and immune evasion in the pathogenesis of cervical cancer [61,67]. Other viral strains such as HPV6, HPV11, and HPV90 were reported in different cancer-related processes like genital warts and cervical intraepithelial lesions in women [68,69]. Although low-risk HPVs are generally considered less involved in carcinogenesis, some studies have identified cases where low-risk HPV presence may be also linked to anal cancer [70]. Additionally, coinfection with low-risk HPV strains, such as HPV6 and HPV18, alongside high-risk HPV types has been reported in anal cancer samples [71]. Building on this knowledge, our in silico analysis reveals that low-risk HPV viral miRNAs may also play a regulatory role in ASCC progression, a role that remains scarcely reported.

The high-confidence set of de novo HPV-miRNAs identified in this study is predicted to target host genes involved in critical tumorigenic pathways associated with the development and progression of ASCC, offering novel insights into host–pathogen regulatory interactions that remain incompletely understood. Supporting these findings, TargetScan predictions based on anti-correlated RNA expression revealed a substantial number of human genes potentially subject to negative regulation by viral miRNAs at more advanced stages of anal lesions. This observation aligns with the increased number of differentially expressed PCGs (FDR < 0.01) identified in comparisons involving ASCC, suggesting that miRNA-mediated disruptions of host transcriptional programs may contribute progressively to neoplastic transformation in ASCC.

Subsequently, in order to investigate the potential functions of these transcriptional profiles, functional enrichment analysis across the different comparisons was performed. When comparing ASCC vs. HGSILs, biological terms related to the cell cycle were identified, such as meiotic nuclear division and the meiotic cell-cycle process, indicating a possible regulation of cell division in the early stages of the lesions. Although the enrichment of meiosis-associated pathways in somatic cells was initially surprising, a closer examination of the genes driving this signal suggests that several may be aberrantly reactivated in cancer contexts—an aspect often overlooked or underrepresented in current Gene Ontology classifications. For instance, the gene *SYCP2L*, a paralog of the meiosis-specific synaptonemal complex protein 2 (*SYCP2*), which is involved in chromosome localization and axis assembly during meiotic recombination [72], was found to be upregulated in ASCC and annotated with the enriched GO terms. Given that *SYCP2* has been reported as aberrantly expressed in ovarian and breast cancers—where its overexpression is associated with resistance to therapies targeting the DNA damage response [73]—it is plausible that *SYCP2L* may exhibit similar aberrant activation in the context of ASCC. While only limited studies have linked SYCP2L to female infertility [74,75], its sequence homology with SYCP2 raises the possibility of shared non-canonical functions, such as involvement in the negative regulation of cell death via DNA repair [73]. This potential role may be relevant in light of the enrichment of meiosis-related pathways observed in the ASCC vs. HGSIL comparison. This raises the possibility that the downregulation of HPV6-miRNA 839-861---NC_075235.1, identified as a high-confidence de novo miRNA in our analysis and putatively targeting SYCP2L expression, may function as an oncoMiR, promoting the progression from HGSIL to ASCC.

An additional functionally enriched term of interest was related to lipid metabolism, which aligns with the existing literature, highlighting the importance of lipid metabolic reprogramming in HPV-driven carcinogenesis. For instance, it has been shown that the expression of the HPV16 E6 oncoprotein upregulates miR-21, which in turn promotes the expression of several enzymes involved in lipid metabolism [76]. This misregulation of lipid metabolism allows HPV to modulate the composition of the cell membrane in order to control infection efficiency, evade the immune response, and manage energy reservoirs for cellular reprogramming in tumor contexts [77]. In our enrichment analysis, lipid metabolism alteration mostly corresponds to the overexpression of the *APOC1* gene in ASCC, which has been also found to be upregulated in HPV-positive cervical cancer [78]. Among the findings derived from the de novo identified miRNAs in this study, we can highlight that the downregulation of HPV6-miRNA 3071-3094-+-NC_075235.1 seems to be highly associated with the upregulation of APOC1 in ASCC.

Additional relevant terms such as the enrichment related to interferon-alpha response pathways suggests a potential modulation of immune signaling during lesion progression. This observation aligns with previous studies reporting that HPV can affect interferon-mediated signaling, disrupt immune cell trafficking and adhesion, and alter cellular phenotypes. These disruptions are known to contribute to immune evasion, creating a microenvironment conducive to persistent infection and malignant transformation.

Similarly, the enrichment of pathways related to polyamine biosynthesis and amino compound metabolism highlights the importance of miRNA-targeted protein-coding genes involved in cell growth and proliferation. Prior research has shown that genes regulating polyamine metabolism are upregulated in HPV-infected cervical squamous cell carcinoma, where their activity may enhance the effects of polyamine oxidation. This interaction is proposed to inhibit apoptosis in keratinocytes carrying potentially oncogenic mutations, thereby promoting the survival and expansion of HPV-infected cells within a tumor-promoting context [76,79].

In the comparison between ASCC and LGSILs, enriched biological pathways associated with the positive regulation of mesenchymal cell proliferation were observed. This mesenchymal phenotype is known to be linked to increased invasiveness and metastasis [80]. It has been shown that the degradation of p-53 in HPV-infected epithelial cells in cervical cancer promotes epithelial–mesenchymal transition [81]. These findings suggest that the downregulated miRNAs in ASCC may potentiate this transition, contributing to enhancing the metastatic capacity of HPV-infected cells. Pathways related to the Wnt signaling pathway and cell polarity were also identified in our analysis, such as the non-canonical Wnt signaling pathway and the regulation of planar polarity, highlighting the importance of regulating cell behavior in terms of direction and spatial organization. The E6 and E7 oncoproteins, which are responsible for the oncogenesis of HPV, have been found to be involved in the regulation of the Wnt/β-catenin pathway [82]. Among the genes that are overexpressed in these pathways of ASCC vs. LGSIL analysis are *GPC2* and *GPC3* [83]. Both proteins have been implicated in the development of various types of tumors, such as hepatocellular carcinoma [84] and colorectal cancer [85]. Remarkably, we highlight the downregulation of the de novo HPV11-miRNA 4456-4476---NC_075235.1, which is predicted to target GPC3, suggesting a potential regulatory function in these processes. Taken together, these results suggest that the HPV-miRNAs identified as targeting genes involved in these functions are involved in a wide variety of biological and cellular processes, many of which are essential for development, cell proliferation, and tissue organization during injury progression.

Finally, the comparison between ASCC and LGSIL + HGSIL samples revealed the enrichment of pathways related to cell division and proliferation, including the regulation of the meiotic cell cycle and mesenchymal cell proliferation, highlighting key processes involved in tumor progression. As observed in the ASCC vs. HGSIL comparison, we obtained enriched meiosis-related pathways, whose genes involved in synaptonemal complex assembly have been associated with the regulation of genome integrity in cancer [86]. In our results, several enriched pathways related to the regulation of the oxidative stress response further highlight the potential involvement of miRNAs in protecting against cellular damage at the ASCC stage. Previous studies have shown that several antioxidant enzymes and detoxification pathways are also consistently linked to HPV-transformed cells, which appear to be well-adapted to highly oxidative environments [87]. Other important pathways include cytidine metabolism, the cytidine-to-uridine editing process, and homocysteine metabolism, reflecting the regulation of metabolic and epigenetic processes critical for genomic stability. It has been described that the APOBEC family of cytidine deaminases, which are involved in the enriched cytidine metabolism pathways identified in our analysis, may contribute to the generation of driver mutations within the helical domain of PIK3CA in HPV-positive cancers, thereby promoting HPV-mediated tumorigenesis [88,89]. Supporting this, our data showed the upregulation of *APOBEC3B* and *APOBEC3F* in ASCC, predicted to be regulated by the de novo HPV11-miRNAs 1892-1907-+-NC_075235.1 and 3071-3094-+-NC_075235.1, respectively. Lastly, we reported some pathways related to the viral life cycle and viral genome replication, such as negative regulation of viral genome replication, suggesting that HPV-miRNAs may also be involved in modulating the cell response to viral infection.

It is important to highlight that functional validation remains a critical next step to confirm the biological relevance of these miRNA–mRNA interactions. In this context, the work of Peronace et al. (2024) [90] provides a compelling example of in vitro validation of miRNA-based biomarkers as clinically useful tools. They highlighted the double methylation of *FAM194* and has-miR-124-2 as a promising tool for diagnosing high-risk HPV-positive women with early-stage cervical lesions. Remarkably, their study reported a frequent association between HPV genotypes 16 and 18 and positive methylation of both genes, further supporting the relevance of these markers in HPV-driven carcinogenesis. Such findings underscore the importance of integrating molecular signatures into screening strategies for HPV-driven neoplasms, including ASCC. To advance beyond computational inference, future studies should incorporate experimental approaches widely used in the validation of viral miRNAs. For instance, the Dual-Luciferase^®^ Reporter Assay has been successfully applied in hepatocellular carcinoma to confirm the regulation of PPM1A by HBV-miR-3 [91]. Additionally, overexpression or knockdown of specific viral miRNAs, combined with transcriptomic profiling (e.g., RNA-seq), could help uncover their broader effects on host gene expression. Techniques such as RNA immunoprecipitation (RIP) and CLIP-seq can identify direct physical interactions between miRNAs and their target transcripts within the RISC complex. High-throughput approaches like HITS-CLIP, PAR-CLIP, and qCLASH have also proven effective in mapping extensive viral miRNA–target networks, as demonstrated in herpesvirus models. The integration of these methodologies will be essential to validate and expand upon the regulatory mechanisms proposed here within the host–virus interaction landscape [36]. The integration of these techniques would contribute to robust validation of the postulated interactions and elucidate the functional impact of viral miRNAs on host gene regulation.

Beyond the biological findings, our study also serves as a proof of concept for the reanalysis of publicly available total RNA-Seq datasets to simultaneously profile host gene expression, viral content, and microbiota composition. This approach aligns with FAIR data principles and demonstrates the potential of leveraging existing sequencing data to uncover novel virus–host regulatory interactions. The pipeline we implemented is standardized and generalizable, offering a valuable resource for future investigations across other datasets, viral species, and HPV-associated diseases. As such, we believe this framework can contribute to advancing the understanding of virus-driven tumorigenesis through integrative, data-driven methodologies.

## 4. Material and Methods

### 4.1. Data Acquisition and Comparative Design

We used the transcriptomics dataset published by Lacunza et al., 2024 (GSE253560, GEO repository) [62]. This dataset comprises 31 LGSIL samples, 16 HGSIL samples, and 23 ASCC samples. ASCC samples correspond to patients with a confirmed cancer diagnosis, whereas LGSIL and HGSIL samples represent pre-invasive lesions of low- and high-grade severity, respectively. Details regarding sample collection, processing, and the RNA-Seq protocol are available in the original publication [62]. We focused our comparative analyses on specific pairwise contrasts, including ASCC versus LGSILs, ASCC versus HGSILs, LGSILs versus HGSILs, and ASCC versus LGSILs + HGSILs, in order to characterize more accurately the changes associated with the progression of pre-invasive lesions to invasive cancer.

### 4.2. Processing, Mapping, and Analysis of Human Transcriptomics Data

The reanalyzerGSE pipeline [92] was employed to conduct sample quality control, align sequences, and identify human differentially expressed genes. Raw reads were aligned using HISAT2 [93] against the PCGs of the *Homo sapiens* Gencode genome build GRCh38-v47 in very-sensitive mode. Once the reads were aligned, gene-level quantification was performed using featureCounts [94], with MOCB parameters applied: excluding multimapping reads, considering only properly paired reads aligned in a concordant manner, and including reads overlapping annotated features. Subsequently, gene expression matrices were generated for each sample, and differential expression analysis was carried out using the edgeR R package (v4.0.1) [95]. To account for differences in library size and sequencing depth across samples, counts were normalized using the trimmed mean of M-values (TMM) method [96], which is implemented by default in edgeR. This normalization ensures that observed differences in gene expression are not driven by technical biases. Multiple-comparisons adjustments were performed using the Benjamini–Hochberg method. The R package clusterProfiler [97] was used to conduct functional enrichment analysis and explore the biological implications of differentially expressed genes. Gene annotation was performed by mapping Entrez gene identifiers using an orgDB R object, generated through the AnnotationForge package, ensuring seamless integration with clusterProfiler for downstream analyses.

Unaligned reads were retained for taxonomic classification and microbial quantification by Kraken/Bracken [98]. The resulting reports were further analyzed to evaluate the microbial composition. To assess differences across conditions, we quantified the number of reads assigned to each taxon and processed them using the metagenomeSeq R package (v2.1.6) [99]. Taxa with a total read count of zero across all samples—typically introduced by the software despite absence—were removed prior to analysis. To correct for sequencing depth variation, read counts were normalized using the Cumulative Sum Scaling (CSS) method, which adjusts for library size while preserving the structure of sparse count data typical of microbiome datasets. This normalization approach mitigates the influence of highly abundant features and allows for more accurate comparisons across samples. Differential abundance analysis was then carried out using the zero-inflated Gaussian (ZIG) model implemented in metagenomeSeq [99].

### 4.3. Analysis Workflow for Viral miRNA Identification

To identify potential viral miRNAs with biological relevance in anal neoplasia, we employed a targeted two-step approach. First, we performed a differential abundance analysis of viral species across the comparisons performed among the ASCC, HGSIL, and LGSIL samples, identifying a subset of viruses whose presence varied significantly between lesion grades (see previous section). This step allowed us to prioritize a biologically relevant set of viral candidates that may be associated with tumor phenotype.

In the second step, we applied vsRNAfinder [100], a computational pipeline optimized for the detection and annotation of small RNAs (sRNAs), to identify putative viral miRNAs encoded by the genomes of the differentially abundant viruses. This tool requires as input raw RNA-Seq data in fastq format and the reference genome of each virus; alignments were performed using Bowtie v1.3.0 under default parameters [101]. Although the tool is capable of detecting various classes of small RNAs, our analysis focused exclusively on the identification of canonical and non-canonical viral miRNAs. For miRNA prediction, vsRNAfinder requires that the expression level within the coding region should be higher compared to its background. To account for this, the background region was defined as 10 nucleotides both upstream and downstream of the coding region. Additionally, a significance threshold of a *p*-value < 0.05 and a length range of 15–34 nucleotides were selected, based on the typical length of functional sRNAs, such as miRNAs and siRNAs, which generally range between 18 and 30 nt [101]. This threshold optimizes the accuracy of the analysis by minimizing background noise and focusing on the detection of true sRNAs, thereby reducing the likelihood of false positives.

As a validation method to ensure that the viral miRNAs detected by vsRNAfinder were not misclassified as human miRNAs, all identified sequences were queried against miRBase [102]. This analysis confirmed that none of the detected miRNAs exhibited full alignment with any known human miRNA, minimizing the likelihood of erroneous classification.

Additionally, to reduce the risk of false positives and ensure biological relevance, we implemented an additional filtering criterion: only miRNAs derived from viral species previously detected as significantly enriched (FDR ≤ 0.05) in the corresponding comparison groups were retained. While we acknowledge that viral miRNAs may still be expressed at low levels or in latent infections, our conservative strategy aimed to prioritize robust and phenotype-associated candidates for downstream analysis.

### 4.4. Prediction of miRNA-Regulated Target Genes

TargetScan (Release 8.0) was used with the list of differentially expressed PCGs to predict potential targets of the viral miRNAs [103]. To minimize the likelihood of false positives, only differentially expressed PCGs with an FDR < 0.01 were retained. Then, we used TargetScan to compare the 3′UTR sequences of differentially expressed genes against the seed sequences of our predicted miRNAs. The seed region, consisting of seven nucleotides spanning positions 2 to 8 of the miRNA sequence, is a critical determinant for miRNA-mRNA binding [104]. The predicted target genes can be categorized based on the degree of complementarity between the miRNA seed region and the 3′UTR sequence of the target genes. Depending on the number of aligned nucleotides, the predictions are classified as 6mer, 7mer, or 8mer matches. This classification offers insight into the binding affinity of a given miRNA to its target mRNA and, consequently, its potential regulatory effect on transcription. At this stage, all three types of binding sites were retained to avoid excluding any potential targets of interest.

Given that one of the primary mechanisms of miRNA-mediated regulation involves transcript degradation or translational repression, miRNAs typically display expression patterns that are inversely correlated with those of their target mRNAs. This negative correlation reflects the functional outcome of miRNA activity, whereby increased levels of an miRNA lead to reduced expression of its target genes. To prioritize biologically meaningful interactions and reduce the number of spurious associations, we limited our analysis to differentially expressed genes (DEGs) with a false discovery rate (FDR) < 0.01. Within this subset, we then assessed miRNA–mRNA expression relationships using the Spearman correlation coefficient, a non-parametric measure that captures monotonic associations without assuming linearity, in addition to applying a statistical significance threshold (*p*-value < 0.05).

## 5. Conclusions

In this study, we developed and implemented a novel in silico pipeline for the reanalysis of publicly available total RNA-seq data, enabling simultaneous profiling of host transcriptomics, microbial abundance, and viral miRNA discovery. This approach highlights the value of applying FAIR (Findable, Accessible, Interoperable, and Reusable) data principles to extract new layers of biological insight from existing datasets and demonstrates its potential applicability to other diseases and contexts.

Our integrative analysis provides a comprehensive characterization of HPV-derived miRNAs potentially involved in the progression of anal squamous cell carcinoma (ASCC). We identified novel viral miRNAs from HPV6, HPV11, and HPV90 with distinct expression patterns across lesion grades. These miRNAs appear to regulate host gene networks implicated in key tumorigenic processes, including cell-cycle control, epithelial–mesenchymal transition, lipid metabolism, immune modulation, and tissue organization.

Taken together, our findings suggest that viral miRNAs—particularly those encoded by low-risk HPV types—may contribute not only to the viral life cycle but also to the modulation of host gene expression and the promotion of neoplastic transformation. These results offer novel hypotheses about virus–host interactions in ASCC and lay the groundwork for future functional validation studies and biomarker discovery in HPV-associated cancers.

## Figures and Tables

**Figure 1 ncrna-11-00043-f001:**
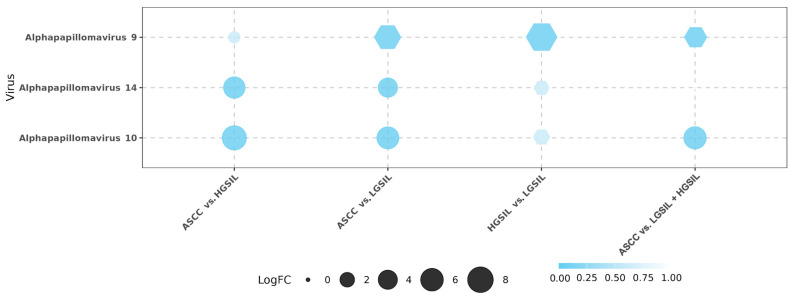
Differential abundance analysis of viruses across the four comparisons: anal squamous cell carcinoma (ASCC) vs. high-grade squamous intraepithelial lesions (HGSILs), ASCC vs. low-grade squamous intraepithelial lesions (LGSILs), HGSILs vs. LGSILs, and ASCC vs. LGSILs and HGSILs. Circles represent downregulated viruses, while hexagons represent upregulated viruses among the samples. The size of each point corresponds to the logFC (fold change) value. False discovery rate (FDR) is represented using a color gradient.

**Figure 2 ncrna-11-00043-f002:**
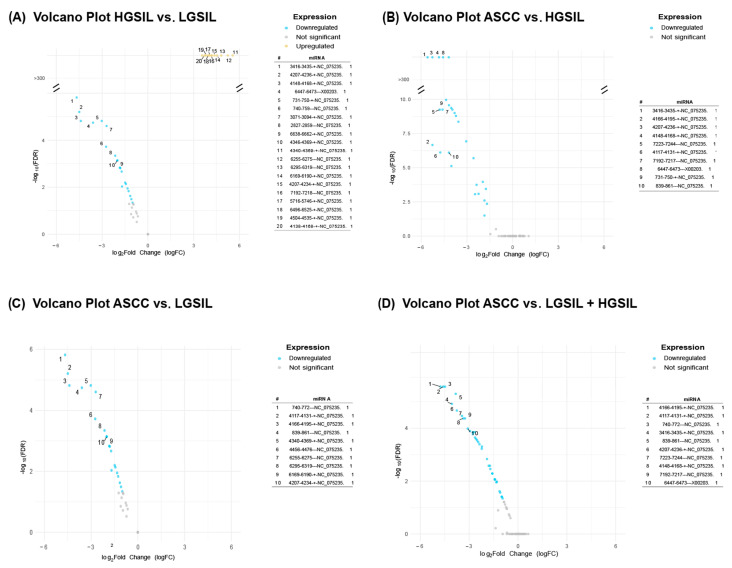
Volcano plots showing differentially expressed HPV miRNAs across the following comparisons: (**A**) high-grade squamous intraepithelial lesions (HGSILs) vs. low-grade squamous intraepithelial lesions (LGSILs), (**B**) anal squamous cell carcinoma (ASCC) vs. HGSILs, (**C**) ASCC vs. LGSILs, and (**D**) ASCC vs. combined HGSILs + LGSILs. Colored dots indicate significant miRNAs (FDR < 0.05), yellow dots represent upregulated miRNAs, and blue dots indicate downregulated ones. For each comparison, the most significantly upregulated or downregulated miRNAs were listed according to their fold changes (|logFC|).

**Figure 3 ncrna-11-00043-f003:**
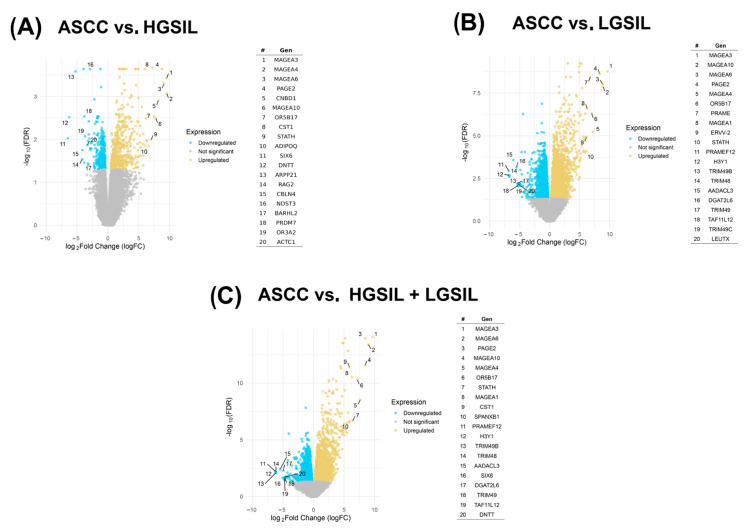
Volcano plots represent differentially expressed protein-coding genes (PCGs) in three comparisons: (**A**) anal squamous cell carcinoma (ASCC) vs. high-grade squamous intraepithelial lesions (HGSILs), (**B**) HGSILs vs. low-grade squamous intraepithelial lesions (LGSILs), and (**C**) ASCC vs. combined HGSILs + LGSILs. Colored dots represent statistically significant miRNAs (FDR < 0.05): yellow points represent upregulated PCGs, while blue points depict downregulated PCGs. Labeled genes are those with the largest absolute logFC (fold change) values and statistical significance. For each comparison, the most significantly upregulated or downregulated genes were listed according to their fold changes (|logFC|).

**Figure 4 ncrna-11-00043-f004:**
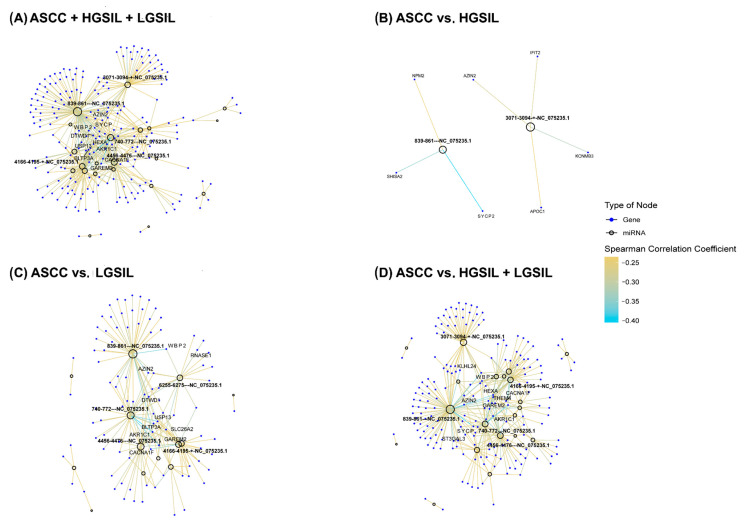
Networks representing the correlation patterns between differentially expressed human papillomavirus (HPV) miRNAs and their significantly anti-correlated target protein-coding genes (PCGs) (*p* < 0.05) across all comparisons: (**A**) anal squamous cell carcinoma (ASCC), high-grade squamous intraepithelial lesions (HGSILs), and low-grade squamous intraepithelial lesions (LGSILs); (**B**) ASCC vs. HGSILs; (**C**) ASCC vs. LGSILs; and (**D**) ASCC vs. HGSILs + LGSILs. Empty circles denote HPV-miRNAs, while blue circles represent target genes. Edges indicate significant negative correlations, calculated using the Spearman correlation coefficient, with edge color intensity corresponding to the strength of the correlation. The ten most highly anti-correlated genes and the five HPV-derived miRNAs with the highest degree of connectivity are labeled in each network. For enhanced visualization and interactive exploration, these networks are shown in HTML versions of the figures, available at http://bioinfo.ipb.csic.es/Papers/ncRNA_HPV/, accessed on 1 June 2025.

**Figure 5 ncrna-11-00043-f005:**
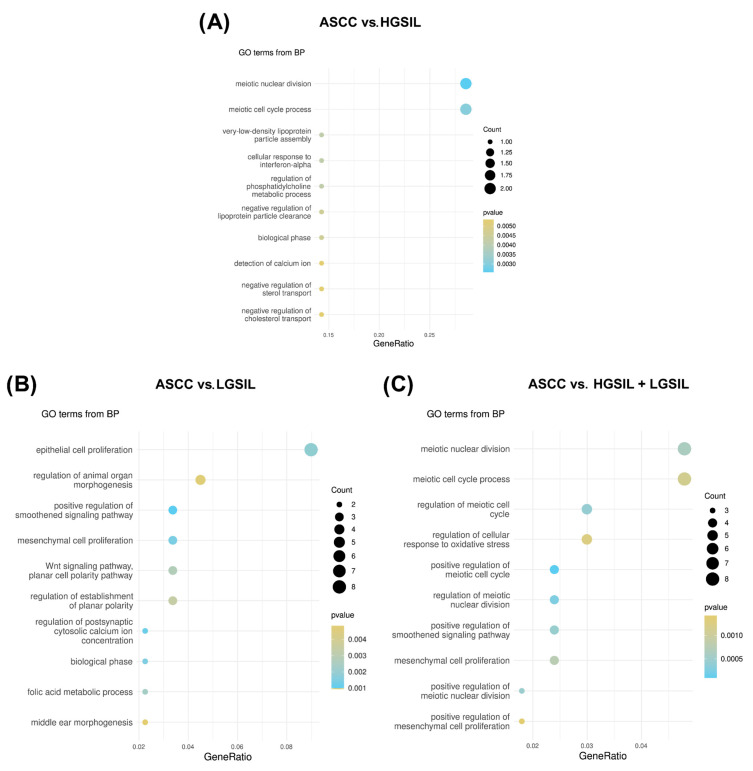
Functional enrichment analysis of differentially expressed PCGs predicted to be target genes regulated by differentially expressed HPV-miRNAs. Ten most significantly enriched GO biological processes pathways ordered by gene ratios in the comparisons (**A**) anal squamous cell carcinoma (ASCC) vs. high-grade squamous intraepithelial lesions (HGSILs), (**B**) ASCC vs. low-grade squamous intraepithelial lesions (LGSILs), and (**C**) ASCC vs. HGSILs + LGSILs are shown. The size of the dots represents the number of predicted target genes in the gene list associated with the GO term and the color of the dots represents the *p*-value.

## Data Availability

The raw data analyzed in this study are available in the Gene Expression Omnibus (GEO) repository (https://www.ncbi.nlm.nih.gov/geo/), reference number GSE253560 (accessed on 21 October 2024). The data derived from the analyses are contained within the article and Appendix A.

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
