# Peer review of "Human Papillomavirus-Encoded microRNAs as Regulators of Human Gene Expression in Anal Squamous Cell Carcinoma: A Meta-Transcriptomics Study"

_ncrna, 2025, doi:10.3390/ncrna11030043_

Round 1

Reviewer 1 Report

Comments and Suggestions for Authors

The manuscript presents a study into the role of viral miRNAs, particularly HPV-derived, in the progression of anal squamous cell carcinoma (ASCC). The integration of metatranscriptomics and de novo viral miRNA discovery is innovative and adds significant value to the field of viral oncology and host-pathogen interaction studies. The analysis is detailed and well-structured, with appropriately curated datasets and rigorous computational pipelines.

However, several areas would benefit from clarification, refinement, and expansion to enhance the manuscript’s clarity, impact, and reproducibility.

Major Revisions:

While the study identifies de novo viral miRNAs, the connection to host gene regulatory effects remains largely predictive. The authors should discuss the limitations of relying solely on in silico predictions and ideally suggest future in vitro validation experiments (e.g., luciferase assays or knockdown studies) to support the postulated interactions.

Overemphasis on Low-Risk HPV Strains
A surprising finding is the significant role ascribed to miRNAs from HPV6, 11, and 90—classified as low-risk types. This needs to be more critically evaluated. Are these results possibly due to higher read coverage for these strains or sampling bias? Including a brief discussion on possible confounders would strengthen the interpretation.

Inconsistencies in Viral Abundance vs. miRNA Expression
The manuscript mentions some miRNAs were excluded due to mismatch with viral abundance in the same comparison. This decision seems arbitrary—why not further explore whether miRNAs from undetected viruses could still exert regulatory effects (e.g., due to latent infection or integration)? Clarify the rationale for exclusion in the Methods section.

Enrichment Analyses and Biological Interpretation
The GO term enrichment is comprehensive but lacks depth in biological interpretation. For instance, the implication of meiotic cell cycle genes in cancer progression should be better contextualized. Why would SYCP2L upregulation be functionally relevant in ASCC, a non-germline tissue?

Figure/Visuals Refinement
Figures 2–5 (volcano plots, networks, enrichment) are helpful, but crowded and at times hard to read. Increase label readability, perhaps by highlighting only top 5 most significant nodes in network graphs or using interactive supplementary files.

To further strengthen the clinical relevance and translational potential of the study, the authors should consider referencing and discussing findings from Peronace et al. (2024). 
Peronace, Cinzia et al. “FAM19A4 and hsa-miR124-2 Double Methylation as Screening for ASC-H- and CIN1 HPV-Positive Women.” Pathogens (Basel, Switzerland) vol. 13,4 312. 11 Apr. 2024, doi:10.3390/pathogens13040312

Typos/Grammar

"Smothered signaling pathway" should likely be "Smoothened signaling pathway" (likely a typo).

Reviewer 2 Report

Comments and Suggestions for Authors

Even though the extensive analysis of the Role of Viral Non-Coding RNAs in Host Gene Regulation and Disease Development has been  published, this article lacks novelty. However, this paper also introduces several novel perspectives.

  1. 1. The title was over-stated and not highly related to the content of the article, for the paper was mainly focused on oncogenic human papillomaviruses.
  2. 2. English must be improved; an extensive revision is needed since errors throughout the manuscript are evident.
  3. 3. The Abstract should be revised as it cannot fully reflect the content of the article.
  4. 4. In introduction part, authors should be provide more reports related to other ncRNA, not only miRNAs.
  5. In results part, most figures are hardly to read, the note-font was too small.

Round 2

Reviewer 1 Report

Comments and Suggestions for Authors

Thanks for your response. I accepted the manuscript in the current form. 

Reviewer 2 Report

Comments and Suggestions for Authors

I have no other concerns.